# Effect of the Architecture of Fiber-Optic Probes Designed for Soluble Solid Content Prediction in Intact Sugar Beet Slices

**DOI:** 10.3390/s19132995

**Published:** 2019-07-07

**Authors:** Ryad Bendoula, Arnaud Ducanchez, Ana Herrero-Langreo, Pablo Guerrero-Castro, Jean-Michel Roger, Alexia Gobrecht

**Affiliations:** 1IRSTEA, UMR ITAP, Montpellier University, 361 Rue J.F. Breton, F-34196 Montpellier CEDEX 5, France; 2Supagro Montpellier, IRSTEA, UMR ITAP, Montpellier University, 2 place Pierre Viala, F-34060 Montpellier, France; 3School of Biosystem and Food Engineering, University College of Dublin (UCD), Belfiled, Dublin 4, Ireland

**Keywords:** sugar beet, NIR spectroscopy, fiber-optic probe

## Abstract

Sugar beet is the second biggest world contributor to sugar production and the only one grown in Europe. One of the main limitations for its competitiveness is the lack of effective tools for assessing sugar content in unprocessed sugar beet roots, especially in breeding programs. In this context, a dedicated near infrared (NIR) fiber-optic probe based approach is proposed. NIR technology is widely used for the estimation of sugar content in vegetable products, while optic fibers allow a wide choice of technical properties and configurations. The objective of this research was to study the best architecture through different technical choices for the estimation of sugar content in intact sugar beet roots. NIR spectral measurements were taken on unprocessed sugar beet samples using two types of geometries, single and multiple fiber-probes. Sugar content estimates were more accurate when using multiple fiber-probes (up to R^2^ = 0.93) due to a lesser disruption of light specular reflection. In turn, on this configuration, the best estimations were observed for the smallest distances between emitting and collecting fibers, reducing the proportion of multiply scattered light in the spectra. Error of prediction (RPD) values of 3.95, 3.27 and 3.09 were obtained for distances between emitting and collecting fibers of 0.6, 1.2 and 1.8 µm respectively. These high RPD values highlight the good predictions capacities of the multi-fiber probes. Finally, this study contributes to a better understanding of the effects of the technical properties of optical fiber-probes on the quality of spectral models. In addition, and beyond this specificity related to sugar beet, these findings could be extended to other turbid media for quantitative optical spectroscopy and eventually to validate considered fiber-optic probe design obtained in this experimental study.

## 1. Introduction

Sugar beet is the second biggest world contributor to sugar production and the only one grown in Europe. Great efforts are currently being made to preserve and improve its competitiveness in an international context marked by increased demand and sugar cane predominance. One lever is to produce new high potential varieties. Within this context, new phenotyping methods are needed to assess the plant characteristics during the breeding program. To be effective, these phenotyping methods have to be non-destructive and rapid to ensure monitoring and high pace. One of the main bottlenecks identified is the lack of effective and suitable tools for assessing sugar content in unprocessed sugar beet roots, directly in field, allowing monitoring growth and yield.

Visible and near infrared spectroscopy (Vis-NIRS) is a well-adapted technology to address this demand, as it is quick, non-destructive, not pollutant and cost effective when compared to standard chemical analysis [1]. It has currently been widely used to assess a wide range of components on fresh fruits and vegetables [2,3,4] and it is especially effective for sugar content estimation [5,6,7].

In the sugar industry, NIRS has been long used to monitor product quality on sugar processing lines in both sugar beet and cane industrial processes [1,8]. However, in the main, studies are based on measurement performed on sugar beet brei samples, processed according to the standard method used in sugar refineries. In the case of homogenous samples, the results achieved for the determination of soluble solid content (SSC) and sucrose prediction from NIR spectra are promising [1]. In a more recent work, Pan et al. [9] studied the feasibility of Vis-NIR spectroscopy for predicting the sucrose content of sliced and intact sugar beets. As expected, better results were obtained on beet slices than on intact beets. Pan et al. [9] have concluded by the need to improve the sensing configuration and calibration methods, in order to overcome the problem of the skin, which is affecting the signal quality and consequently the calibration model performance. One solution is to measure the diffuse reflectance, directly in contact with the flesh, using an invasive but non-destructive fiber-optic probe.

Fiber-optic probes have a double role in these optical systems: (i) delivery of illumination to the target and (ii) collection and delivery of the signal to the detector [10]. Indeed, fiber-optic probes are simple and inexpensive to make, small in size, and can be used in endoscopic or minimally invasive procedures, like, for example, in biomedical applications [11,12,13,14]. The main advantage is that the design of the probe can be adapted to the measurement context.

The use of optical fiber probes coupled with VIS-NIR spectroscopy has already proved for food quality [15]. This NIR method was used to determine the internal quality of fruits [16]. In particular, soluble solid content in intact kiwifruit [17], peaches [18], strawberries [19] and citrus fruits [20] were investigated. In these different works, an “interactance” probe was used and consisted of an inner ring receptor (central bundle of fiber-optics) surrounded by a concentric outer ring of fibers illuminator with an outside diameter of about 20 mm. In our application, an invasive probe could measure a spectroscopic signal on the beet root flesh while the root is still in the ground with the smallest fiber-optic probe possible, a few millimeters in diameter ideally, to limit the damage. Although the use of this “interactance” probe has shown good results [17,18,19,20], it is not suitable for invasive measurements given its dimensions.

Within these specifications, the preliminary work is to define the optimal architecture of the optical probe, in order to acquire the best possible signal, as signal quality is one of the main factors influencing the prediction model based on these signals. Different architectures of fiber-optic probes are possible: either the same fiber delivers and collects the light, as in so called single fiber probe (SFP) [21,22] or two separate fibers are used, i.e., for the illumination and for the collection. These so called multiple fiber probe (MFP) are the one classically implemented in diffuse reflectance spectroscopy devices. The SFP present the advantage to reduce the overall probe diameter, but the fact that in case of contact probes the illumination area and collection area are the same, the signal quality is highly impacted. On the contrary, the distance between the two fibers can be adjusted in case of MPF but it is necessary to find the optimal distance to collect sufficient information from the flesh with a reduced amount of background signal.

In this work, the different optical architectures of a contact probe were tested by measuring Vis-NIR spectroscopic signals of intact sugar beet root slices. The objective was to assess the effect of the probe’s design (i.e., multiple or single fiber mode and distance between illumination and collection point) on the estimation of soluble solid content (SSC) of the sugar beet slice. The comparison was based on both the quality of the signals measured with the different probes and the quality of the prediction models built from these signals. The overall objective was to define the optimal design for a future invasive but non-destructive probe able to measure SSC directly in the field.

## 2. Materials and Methods

### 2.1. Probe Design

In total, three different measurement setups have been designed for this experiment based on three types of geometries:-An integrating sphere (IS), aiming at acquiring reference spectral signatures on the sugar beet slices (Figure 1).-A single fiber probe (SFP), where the same fiber is used for illumination and signal collection (Figure 2a).-A multiple fiber probe (MFP), for which separate fibers are used for light emission and collection and where three different distances between illumination and collection point have been tested: 600 µm, 1200 µm and 1800 µm (Figure 2b).

All the geometries used the same light source, a white light from tungsten-halogen lamp (Ocean Optics HL-200-FHSA, Orlando, Winter Park, FL, USA) and the same spectrometer, a LabSpec4 (ASD, Boulder, Orsay, France) featuring a detection range of 500 nm–2000 nm, with 10 nm resolution.

For the IS, the light source was coupled with a 550 µm core diameter optical fiber (Numerical Aperture (N.A) 0.22, Sedi and ATI, Courcouronnes, France). The light delivered by the fiber was collimated by an aspheric lens of 7.2 mm diameter (F220SMA-C, Thorlabs, Newton, NJ, USA) into an integrating sphere with an active diameter of 30 mm (AvaSphere-50, Avantes, Apeldoorn, Netherlands). The reflected light was coupled inside an optical fiber (0.22 NA, Sedi and ATI) by an aspheric lens of 7.2 mm diameter (F220SMA-C, Thorlabs). This fiber was connected to the spectrometer.

For the SFP, the light source was coupled with a 550 µm core diameter optical emitting-and-collecting fiber (numerical aperture (N.A) 0.22, Sedi and ATI). Reflected light from the sample was coupled inside the emitting-and-collecting fiber which was then connected to the spectrometer.

For the MFP, three illumination-collection fiber separation distances were tested. The light source was coupled with a 550 µm core diameter optical emitting fiber (N.A 0.22, Sedi and ATI). Reflected light from the sample was coupled inside the three-collecting fibers (550 µm core diameter, NA 0.22, Sedi and ATI). These fibers were coupled to an optical switch (DART401, Yenesta) and the optical switch was connected to the spectrometer. The Dart 401 is an optical switch module. This optical switch present 1 × 4 configuration: one common output channel and four input channels. The insertion loss in the fiber, for the whole wavelength range, is less than 1.5 dB. This loss is the same for all the channels. For the multiple-fiber probe, the collecting fibers were located at center-to-center source-collecting distances of 0.6 mm (C_1_), 1.2 mm (C_2_) and 1.8 mm (C_3_).

### 2.2. Sample Preparation

A set of 44 sugar beets composed of 17 different varieties, with a majority of Beta vulgaris var. ‘Acacia’ and Beta vulgaris var. ‘Belladonna’ was collected in the field at different maturity stages (underripe, ripe and over ripe) in order to embrace a large SSC range. For each root, 4 slices of 12 mm thickness were cut perpendicularly to the central axis of the root. Each slice was the cut in three cubic samples of 12 mm side, dedicated to be measured by one of the three optical setups. In total, each optical setup measured 176 (44 × 4) intact flesh samples.

### 2.3. Spectral Acquisition

For each sample, the intensity of the reflected light (*I_s_(λ)*) was measured. Dark current (*I_b_(λ)*) was recorded from all measured spectra and subtracted. A white reference (SRS99, Spectralon^®^) was used as a reference (*I*_0_(*λ*)) to standardize spectra from non-uniformities of all components of the instrumentation (light source, fibers, integrating sphere, spectrometer and optical switch).

From these measurements, a reflectance spectra (*R_s_(λ)*) was calculated for each sample:(1)$$R_{s}\left(\lambda \right)=\frac{I_{s}\left(\lambda \right)-I_{b}\left(\lambda \right)}{I_{0}\left(\lambda \right)-I_{b}\left(\lambda \right)}$$

### 2.4. Soluble Solids Content

Immediately after spectrum acquisition, the samples were placed in individual plastic bags to be stored for seven days at −18 °C prior to SSC measurements. Freezing the samples improved the repeatability of reference measurements, as soluble solids are stored in vacuoles within beet root cells and freezing breaks this compartmentalization. Samples were unfrozen one hour before SSC measurements. SSC was measured through pressing each beet root sample with a garlic press and dropping the juice directly on a digital refractometer (Euromex RD 645, Arnhem, Netherlands). Brix value was used in this study because it is the easiest, least expensive quality parameter to be measured with little preparation. Moreover, the correlation between brix and SSC has been reported by Staunton et al. [23] and Pan et al. [9].

### 2.5. Multivariate Analysis

All computations and multivariate data analysis were performed with Matlab software v. R2012b (The Mathworks Inc., Natick, MA, USA).

### 2.6. Spectral Pretreatments

Models were developed from absorbance spectra that were computed from reflectance spectra measurements (−log R). Data preprocessing methods were tested in order to remove undesired effects like multi-collinearity and the baseline offset on the absorbance spectra: standard normal variate (SNV), Detrend as well as first and second derivatives of the spectra using Savitsky–Golay (SG) smoothing and differentiation. Moreover, several model parameters (number of latent variables and optical spectral range) were evaluated to improve the performance of PLSr models.

### 2.7. Model Calibration

Partial least square (PLS) algorithm [24] was used to calibrate the SSC estimation of the sugar beet cubes. Two thirds of the samples (116 samples) were used for building the calibration model and defining the model parameters. Several model parameters (spectral preprocessing, optimal wavelength range, optimal number of latent variables) were evaluated on the basis figures of merit mainly transcribed by the coefficient of determination (R^2^) and standard error of cross validation (SECV) by leave-one-out method. The second independent data set, composed by one third of the samples (60 samples), was used to test the final calibrated model with the chosen model parameters. Model results were evaluated on the basis of the coefficient of determination (R^2^), the standard error of prediction corrected from the bias (SEPc) and the ratio of sample standard deviation to standard error of prediction (RPD).

## 3. Results and Discussion

### 3.1. Spectral Analysis

Figure 3 shows the reflectance spectra on the different sugar beet flesh cubes with the three different optical setups: integration sphere (Figure 3a), single fiber probe (Figure 3b) and the three positions of multiple fiber probe (Figure 3c–e).

The spectra measured with the integration sphere (IS) was considered here as the reference method that presented a high signal to noise ratio (SNR) and had shapes consistent with what is generally found in NIR spectroscopy of fruits and vegetables [2]: NIR spectra were dominated by water. Two water-related absorption bands can be identified: 980 nm and 1450 nm. Little energy was detected for the region of 1400–1800 nm. This phenomenon was the result of the combination of the increase of the optical path length of the photons in the sugar beet and the strong absorption of light by the beet tissues, due to the combination of the first overtones of bond C–H and O–H in H_2_O. These bands were extensively referenced in other solid vegetable samples [2], especially in those with high water content, as tomato [25] or citrus fruits [26]. In fruit and vegetable products, the absorption bands at 1200 nm have been associated with sucrose [27]. As observed by other authors on solid samples of sugarcane [28], no obvious difference could be seen in the shape of the spectra for different SSC values. A visible base line shift effect was observed in absorbance raw spectra. The increase in the optical path length traveled by the photons in a scattering medium reflects a multiplicative effect on the spectra [29] resulting in baseline drifts of the ideal absorbance spectra.

The second optical design tested was implementing a single fiber probe (SFP), whose measured spectra are shown in Figure 2b. Obviously, these reflectance signals did not reveal the usual absorbing features found in the beet root flesh spectra. The reflectance values were very high, from 65 to 97% and flat over the whole studied wavelength range. In addition, an interference pattern, like waves, is observed. This effect can be due to a thin air wedge between the end of the fiber and the sample. If this air wedge is of the order of a millimeter or less, interferences occur between the waves reflected from end of the fiber and bottom of the sample. This can happen when fiber end is not completely in contact with the sample while measuring. Moreover, in this configuration, the illumination and the collection cones are overlapped. Hence, collected light includes the diffusely reflected light, which has penetrated the sample before exiting, but also the specularly reflected light that seems predominant given the flat absorbance spectra with no visible absorption features.

The last optical architecture designed is implementing a MFP. As showed by the spectra plotted in Figure 3c–e, and whatever the distance between the center of the illumination fiber and the center of the collecting fibers, the spectral features are significantly similar to the spectra collected with the integration sphere. The main difference between the three configurations is the reflectance value of the signal collected with a maximum of 60%, 15% and 8% for multiple fiber probe with a distance of 600 µm, 1200 µm and 1800 µm respectively. Hence, the longer the optical path, the lower the light intensity collected, as the probability of being absorbed increases.

Amelink et al. [30] proposed an expression which decomposes the total signal collected by a contact probe into different types of signals related to the way the light travels in the sample. This expression can be simplified as:(2)$$I\left(\lambda \right)=\left(I_{sp}\left(\lambda \right)+I_{ss}\left(\lambda \right)+I_{ms}\left(\lambda \right)\right)\mathrm{TL},$$
with *I_sp_*(*λ*) the intensity of specularly reflected light at the probe–sample interface, *I_ss_*(*λ*) the intensity of singly (or weakly) scattered light that has traveled a relatively small distance from the fiber tip picked up by the emitting and collecting fiber and *I_ms_*(λ) is the intensity of multiply scattered light picked up by the emitting and collecting fiber. T is the combined transmission function of the fiber probe and the spectrometer channel. Finally, L is the light distribution at the probe-sample interface.
(3)$$I\left(\lambda \right)=\left(I_{ss}\left(\lambda \right)+I_{ms}\left(\lambda \right)\right)\mathrm{TL}.$$

The collected light has traveled in the flesh and therefore becomes more informative as the signal is freed from useless information. This leads to a clearer identification of the chemical information in the spectra. The difference between the three MFP configurations (comprising three different distances between the illumination and collecting fiber cores) impact the proportional contribution of *I_ms_*(*λ*) and *I_ss_*(*λ*) to the total signal. The longer the path, the higher the probability to have mainly multi-scattered light, and inversely. To conclude, the spectra acquired with the multiple fiber probes are, from a spectral point of view, of good quality and contain chemically related information. However, the spectral analysis is not sufficient to assess the potential of the different designs to predict SSC. It is therefore necessary to resort to the establishment of models.

### 3.2. Calibration Models

Soluble solid content (SSC, brix) of the different sample sets per probe, obtained from the digital refractometer are summarized in Table 1.

The SSC range obtained was larger than in other studies because some of the beet roots were kept in soil to overripe them. However, the number of samples with a SSC higher than 40° brix was about 10% of the whole sample set, leading to a large standard deviation and a relatively high skewness.

Calibration and prediction results of SSC for the beet root samples using the different optical setups are presented in Table 2. In this table, column 2 indicates the optimal spectral pretreatments applied on reflectance spectra for each optical configuration to remove undesired effects as multi-collinearity and the baseline offset resulting from scattering effects [9]. Columns 3 and 4 specify the model parameters retained for latent variables and the restricted spectral range respectively which resulted in the best PLS model. Columns 5 to 9 present the classic statistical parameters, including coefficient of determination (R^2^) for calibration and prediction, standard error for calibration (SEC), standard error for prediction (SEP) and the ratio of sample standard deviation to standard error of prediction (RPD). All these parameters were used to assess the performance of each calibration model for predicting the SSC of beet roots.

First of all, the collected reflectance spectra have been log transformed into absorbance spectra. The model built from the measurement using the integrating sphere was of good quality and in the range of model performances of other work, however, presenting a higher SEP [9]. The high SEP was a direct consequence of the lognormal distribution of the brix values of the sample set [27].

Compared to this reference acquisition method, the model built from the SFP measurements showed a relatively similar performance level by comparing the R^2^, SEP and the RPD of the prediction set. However, the number of latent variable was very important (15 LV). Observing the shape of the spectra collected with the SFP (Figure 3b), this high model complexity was expected. Indeed, this confirms that the useful information related to SSC is hidden by the specular reflection. But, despite all the specular reflection collected by the SFP, there is also relevant information that has been collected. Canpolat and Mourant [14] showed that single-fiber measurement geometry maximizes the probability of collecting single scattered photons from small depths.

The models built with the spectra collected using the multiple-fiber probes (whatever the distance between illuminating and collecting fibers), outperform the latter. By separating the illumination fiber and the collecting fiber, all the collected light has travelled a certain distance in the beet-root flesh, and therefore carries more information related to the chemical composition. On the other hand, with this design, the collecting fiber cannot collect specularly reflected light.

If the multiple fiber architecture offers the most informative signals, there is a difference between the three positions tested here. Among the three architectures, the smaller the distance between the emitting and collecting fibers is, the higher is the model performance.

For the larger distance (1800 µm), the measured signal contains mainly photons having travelled a long distance and being multiple scattered. Thus, this signal was affected by the physical structure, enhancing non-linearities between absorbance and absorber’s concentration and baseline drifts. These spectral disturbances directly affect the performance of the PLS model [31,32]. For the two first positions, the calibration and prediction models showed a greater correlation coefficient, higher than 0.9. These high values of R^2^ indicate that the measured signals are strongly correlated to SSC contained in the sugar beets. However, for the first position, the SEP is much lower (1.68 brix) and is getting closer to the SEP observed in other works. The calculated RPD for the positions C_1_ and C_2_ were respectively equal to 3.95 and 3.27. These high values of RPD highlighted the good predictions capacities of the multi-fiber probes, especially for the position C_1_. This relatively small distance between the fibers is the best trade-off for collecting sufficiently enough signals having interacted with the root but with no specular reflection and a small contribution of multiply scattered light. Indeed, the first MFP architecture allows collecting a bigger proportion of single scattered light, less affected by multiplicative effects and baseline drifts.

## 4. Conclusions

We have assessed the potential of probe designs using optical fibers able to measure soluble solid content of sugar beets. Two different fiber-based probe designs have been setup: a SFP in which the same fiber is used for light delivery and collection and MFP where the illumination and the collection fiber are separated. The potential of the different setups was evaluated first throughout the quality of the spectral signatures collected. Indeed, the quality of the signal is highly affected by the architecture of the probe. The single fiber probe allows collecting a signal that has interacted with the sample, but happens to predominantly collect specular reflection, which is overlapping the useful information. Thus, the resulting model present relatively good figures of merit (R^2^ = 0.88 and SEP = 2.34), however, with a high complexity (15 latent variables).

The probes built with a multiple fiber architecture proved to have a higher potential: first the quality of the spectral signature improved, mainly because this architecture allows discarding the specular reflection, and second, the resulting models outperformed all the other setups (SFP and integration sphere). Moreover, the distance between the illumination point and the collection surface also influences the setup performance. It appears that collecting light that has a relatively small path is more efficient. The signal which travelled a too long distance in the sample is distorted by multi-scattering events.

These studied architectures were compatible with the idea to develop a “plug and play” invasive probe of a diameter smaller than 1.5 mm, easy to implement to a commercial spectrophotometer. This invasive but not destructive probe would provide the ability to assess sugar content on intact sugar beets, while they are still in the soil. This is a first mandatory step for quick in-field assessment, which is one of the main current bottlenecks on varietal selection in breeding programs. Finally, beyond this specific study on sugar beet, it would be interesting to investigate other turbid media, as biological tissues, for example, or just on standardized phantom samples, coupled with quantitative optical spectroscopy measurements in order to validate considered fiber-optic probe design obtained in this work and thus to expand the scope of possible applications in other study fields.

## Figures and Tables

**Figure 1 sensors-19-02995-f001:**
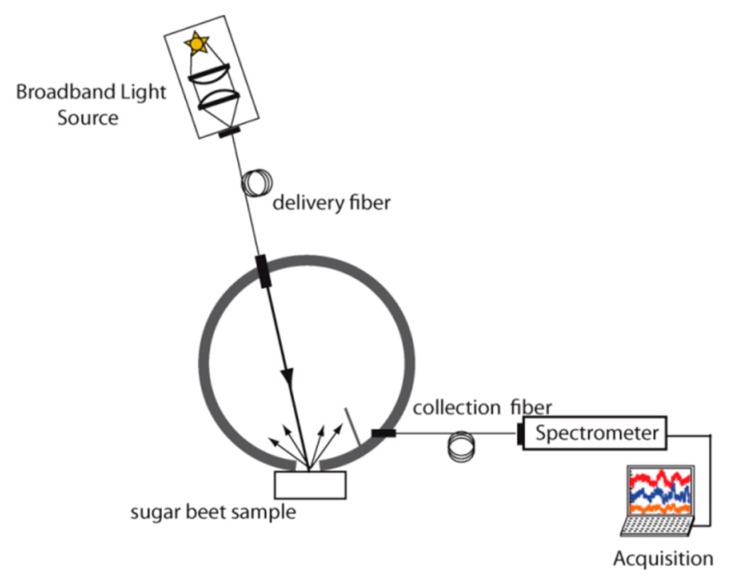
Schematic diagram of integration sphere setup.

**Figure 2 sensors-19-02995-f002:**
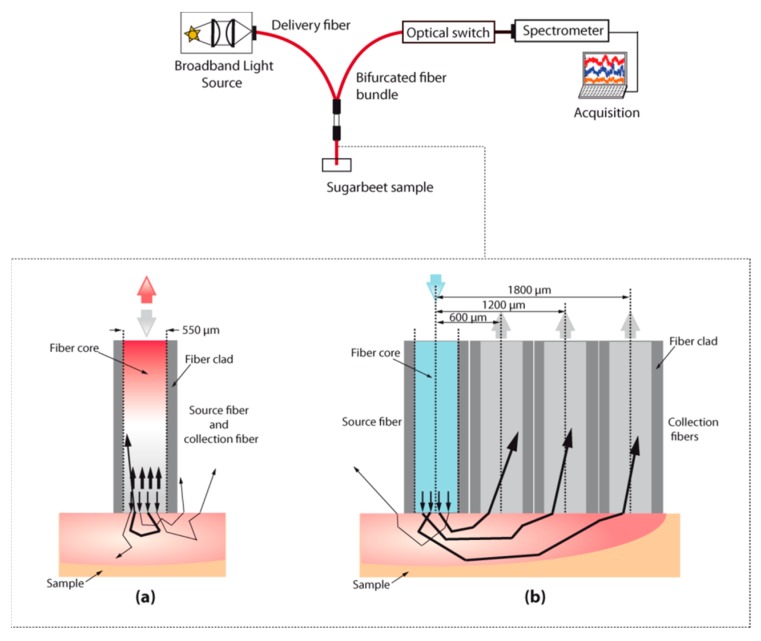
(**a**) Schematic diagram of the single-fiber probe. (**b**) Schematic diagram of the multiple-fiber probes.

**Figure 3 sensors-19-02995-f003:**
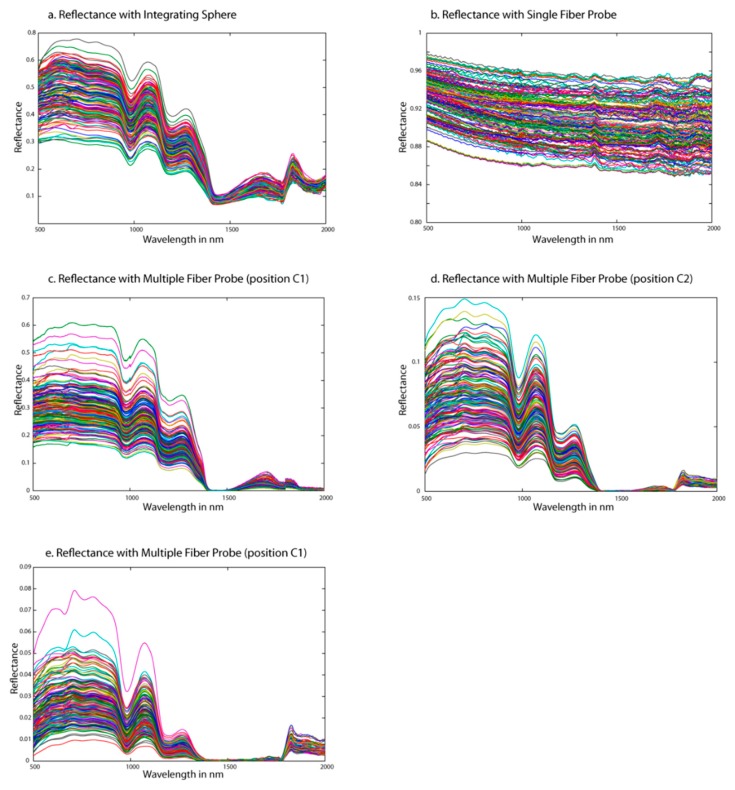
Raw reflectance spectra measured with integration sphere (**a**), single-fiber probe (**b**) and a multiple fiber probe respectively with a distance of 600 µm (**c**),1200 µm (**d**) and 1800 µm (**e**).

**Table 1 sensors-19-02995-t001:** Descriptive statistics of soluble solid content values for the different optical designs.

Optical Configurations	Number of Samples	Soluble Solid Content Range (brix)	Soluble Solid Content Mean Value (brix)	Standard Deviation (brix)	Skewness
Integration sphere (IS)	174	[17.3–44.1]	24.22	6.69	1.69
Single-fiber probe (SFP)	175	[17.1–43.2]	24.67	6.96	1.58
Mutiple-fiber probes (all positions)	169	[17.1–45.5]	24.42	6.65	1.70

**Table 2 sensors-19-02995-t002:** Figure of merit of the calibration and prediction using the different optical designs.

Optical Configuration	Spectral Preprocessing	Latent Variables	Spectral Range	Calibration Set		Prediction Set		RPD
				R^2^	SEC	R^2^	SEP	
Integrating sphere	−log, smoothing	8	700–1600 nm	0.906	2.03	0.885	2.39	2.8
Single Fiber Probe	−log	15	500–1800 nm	0.799	3.2	0.883	2.45	2.84
Multiple Fiber Probe C1 (0.6 cm)	−log	9	800–1400 nm	0.935	1.7	0.932	1.68	3.95
Multiple Fiber Probe C2 (1.2 cm)	−log smoothing	10	800–1380 nm	0.936	1.68	0.909	2.03	3.27
Multiple Fiber ProbeC3 (1.8 cm)	−log smoothing	7	500–1344 nm	0.885	2.54	0.896	2.15	3.09

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
