# Peer review of "Effect of the Architecture of Fiber-Optic Probes Designed for Soluble Solid Content Prediction in Intact Sugar Beet Slices"

_sensors, 2019, doi:10.3390/s19132995_

Round 1

Reviewer 1 Report

Please refer to the detailed comments in the attached file.

Author Response

Point 1. Under the introduction section, authors should include some of the recent publications on fiber-optic based sensing devices for measuring the soluble solid content.

Response 1 :

Following your recommendations, some publications were added in the article, lines 64 to 69.

Point 2. In Fig. 2(b), the absorbance signal is not revealing the usual absorbing features found in the beet root flesh spectra. And also, the reviewer understands its due to a thin air wedge amid the fiber-end and sample. This can happen when fiber end is not completely in contact with the sample while measuring. The reviewer suggests that this can be more effective by comparing with the uninformative variable elimination (UVE) method for similar prediction accuracy.

Response 2 :

For Fig. 3(b) (and not Fig. 2(b)), we think that the use of UVE method is not necessary and will not improve the prediction accuracy because it contains mainly, and perhaps even exclusively specular reflection (absorbance spectra are flat over the whole wavelength range with very low values) which is the part of the light that contains no analytic information.

Point 3. The author can also mention the values of RPD for each configuration in the abstract.

Response 3 :

RPD values for multi-fiber probes were added in the abstract.

Reviewer 2 Report

The title of the paper, as well as the motivation and conclusions, are based in a beetroot sugar application. However, the beetroot sugar is secondary for the goals of this work. Authors should review the title and the scope of the work.

Graphs must have units for absorbance and all the curves must fit within the plot area (change the y scale for graphs on fig. 3c, 3d and 3e).

The authors are evaluating methods to improve the detection on NIR spectrometry on turbid media. However, the tests were made on beetroot sugar, which is only a particular case on turbid medium samples. When trying to validate a method, one should stick to standards and then apply on the actual subject. Henceforth, a more controlled experiment is needed to validate the spectra presented, which must include standard targets.

Although the subject is of interest of the food industry, if the objective of the authors is to propose this NIR probing method as a reference method to the characterization of sugar content, a more detailed study must be done. As an example, there is the work done by Papaioannou et al: “Effects of fiber-optic probe design and probe-to-target distance on diffuse reflectance measurements of turbid media:  an experimental and computational study at 337 nm”.

Author Response

Dear reviewer,

Thank you for all comments concerning our article.

The answers to your suggestions below:

Point 1 : The title of the paper, as well as the motivation and conclusions, are based in a beetroot sugar application. However, the beetroot sugar is secondary for the goals of this work. Authors should review the title and the scope of the work.

Response 1 :

In this article, the study object for which the project was founded is sugar beet and one of the expected outcomes of this project is the development of a dedicated tool to measure the sugar content in this specific plant material. Hence, sugar beet is the central element of this study and not secondary for the goals of this work. It is therefore important for us that the word "sugar beet" appears in the title in particular.

Point 2: Graphs must have units for absorbance and all the curves must fit within the plot area (change the y scale for graphs on fig. 3c, 3d and 3e).

Response 2:

The y scale for graph on fig. 3c, 3d and 3e is limited to a value of 8 in absorbance, because beyond, the signal saturates, corresponding to the minimum reflectance. For information and for better understanding, I have attached a pdf file relating to reflectance spectra.

Point 3: The authors are evaluating methods to improve the detection on NIR spectrometry on turbid media. However, the tests were made on beetroot sugar, which is only a particular case on turbid medium samples. When trying to validate a method, one should stick to standards and then apply on the actual subject. Henceforth, a more controlled experiment is needed to validate the spectra presented, which must include standard targets.

Response 3:

Having no information on the physical and chemical parameters of the beet flesh such as the absorption and diffusion coefficients, it is difficult to know on which standard target and/or model media it is necessary to perform the tests and experiments. The results could have been completely different between these standard targets/model media and our study object, the sugar beet. And therefore in the end, completely different probe designs. Hence, and as sugar beet is the central element in this study, as indicated in "Response 1", it was more easier and useful for us to test on sugar beet directly.

Point 4: Although the subject is of interest of the food industry, if the objective of the authors is to propose this NIR probing method as a reference method to the characterization of sugar content, a more detailed study must be done. As an example, there is the work done by Papaioannou et al: “Effects of fiber-optic probe design and probe-to-target distance on diffuse reflectance measurements of turbid media:  an experimental and computational study at 337 nm”.

Response 4:

Indeed, the objective of our work in this article was to propose a NIR probing configuration as a reference method to the characterization of sugar content but dedicated to sugar beet exclusively. The optimal configuration found in this article can not be suitable for other objects of study (other fruits or vegetables) which presents another type of "turbid medium".

Reviewer 3 Report

In this paper the authors presented several fiber optic sensors dedicated for the assessment of the sugar content in  sugar beets roots. The topic of this research is important especially that the method showed by authors can be used even if the root is still in the soil.

Generally,  I have found the study as interesting and I recommend it for publication.

1.         I haven’t noticed table 1. Please check tables numerations.
2.         Please explain better what exactly is presented in Table 2 column Soluble Solid Content. Are values shown in this column  obtained for one sample? or for set of samples(average value)
3. Please describe Table 3 in the text.

Author Response

1. I haven’t noticed table 1. Please check tables numerations.

We agree with the reviewer, there was a problem with tables numerations. We rectified this numeration problem in the text:

“Table 2” replaced by “Table 1”

“Table 3” replaced by “Table 2”

2. Please explain better what exactly is presented in Table 2 column Soluble Solid Content. Are values shown in this column obtained for one sample? or for set of samples (average value).

This table describes the statistical parameters of the Soluble Solid Content values of the sample sets used for each optical probe configuration. This column represents the SSC mean values of each sample sets. This allowed to check if the sample sets used for each optical configurations were similar in order to avoid generating a bias.

3. Please describe Table 3 in the text.

We added some sentences in the text as suggested: from line 288 to 296.

Round 2

Reviewer 2 Report

None of the issues pointed out on the first review were satisfactorily addressed.

Author Response

Reviewer 2:

Point 1: The title of the paper, as well as the motivation and conclusions, are based in a beetroot sugar application. However, the beetroot sugar is secondary for the goals of this work. Authors should review the title and the scope of the work.

Response 1:

We agree with the reviewer; the scope of this work extends far beyond the application on sugar beet. Nevertheless, this article is the result of a project on the characterization of the quality of sugar beet. As mentioned previously, one of the expected outcomes of this project is the development of a dedicated tool to measure the sugar content in this specific plant material. Thus, sugar beet is a key element of this study and we must keep this word in the title. However, we have added a few sentences to broaden the scope of this article, in the abstract (from line 31 to 34) and conclusion (from line 354 to 358).

Point 2: Graphs must have units for absorbance and all the curves must fit within the plot area (change the y scale for graphs on fig. 3c, 3d and 3e).

Response 2:

Absorbance spectra have been replaced by reflectance spectra for better readability and comprehension. Hence, some lines in the text in part 3.1 “Spectral analysis” have been changed.

Point 3: The authors are evaluating methods to improve the detection on NIR spectrometry on turbid media. However, the tests were made on beetroot sugar, which is only a particular case on turbid medium samples. When trying to validate a method, one should stick to standards and then apply on the actual subject. Henceforth, a more controlled experiment is needed to validate the spectra presented, which must include standard targets.

Response 3:

We agree with the reviewer that the conclusions done on sugar beets must be confirmed on other materials and furthermore on standard samples. This work will be done soon and will give rise to another paper. Some text has been added in this sense in the conclusion (from line 354 to 358). After, as mentioned previously, there is a real difficulty to know and to choose on which standard or model media, that would approach the physical and chemical internal parameters of sugar beet because we have no information of this properties, could be used to perform the experiments to validate this method and thus this probe design. The identified risk is that the results could be different, see completely opposite, between these standard targets/model media and our study object (sugar beet) which leads to radically different probe designs. Thus, it seemed more adapted and useful to proceed with the optimization of the probe design based directly on sugar beet.

Point 4: Although the subject is of interest of the food industry, if the objective of the authors is to propose this NIR probing method as a reference method to the characterization of sugar content, a more detailed study must be done. As an example, there is the work done by Papaioannou et al: “Effects of fiber-optic probe design and probe-to-target distance on diffuse reflectance measurements of turbid media: an experimental and computational study at 337 nm”.

Response 4:

The answer to this question is along the same line as in question 3. The objective of our work in this article was to propose an optical probe design dedicated to the characterization of sugar content to sugar beet exclusively. It’s clear that optimal configuration found in this study cannot be suitable for other study objects (other fruits or vegetables, biological tissues…etc) which presents another type of turbid media with different properties. By contrast, if indeed we had considered our probe as a reference method to the characterization of sugar content for any other media, a more detailed study with other tests and experiments should have been performed to validate our method and probe design but is not the message and the main objective in this study. This work with a more general issue will be done soon and will give rise to another paper.